

# Drug therapy and other factors associated with the development of acute kidney injury in critically ill patients: a cross-sectional study

Danielly Botelho Soares[1], Juliana Vaz de Melo Mambrini[2], Gabriela Rebouças Botelho[1], Flávia Fialho Girundi[1], Fernando Antonio Botoni[3,4] and Maria Auxiliadora Parreiras Martins[1,3,4]

[1] Faculdade de Farmácia, Universidade Federal de Minas Gerais, Belo Horizonte, Minas Gerais, Brazil
[2] Centro de Pesquisa René Rachou, Fundação Oswaldo Cruz, Belo Horizonte, Minas Gerais, Brazil
[3] Faculdade de Medicina, Universidade Federal de Minas Gerais, Belo Horizonte, Minas Gerais, Brazil
[4] Hospital Risoleta Tolentino Neves, Belo Horizonte, Minas Gerais, Brazil

Corresponding author
Maria Auxiliadora Parreiras Martins,
auxiliadorapmartins@hotmail.com

## ABSTRACT

**Background**. Acute kidney injury (AKI) is associated with a significant increase in morbidity, mortality, and health care costs. In intensive care units (ICU), AKI is commonly multifactorial and frequently involves diverse factors, such as hypovolemia, sepsis, and the use of nephrotoxic drugs. We aimed to investigate drug therapy and other factors associated with the development of AKI in a Brazilian public hospital.

**Methods**. This is a cross-sectional study involving critically ill patients at an ICU of a tertiary hospital. All data on sequential serum creatinine ($S_{Cr}$) level, glomerular filtration rate (GFR), and urine output were collected during ICU stay. The primary outcome was the occurrence of AKI assessed by the Acute Kidney Injury Network (AKIN) criterion. Sociodemographics, clinical data and drug therapy were considered as covariates. Factors associated with AKI were assessed using logistic regression.

**Results**. Overall, 122 participants were included in the study. Median age was 46.0 (interquartile range, IQ = 29.0–69.0) years, with a predominance of men (58.2%). Mean number of prescribed drugs throughout ICU stay was 22.0 ± 9.4. The number of potentially nephrotoxic drugs ranged from two to 24 per patient. A total of 29 (23.8%) ICU patients developed AKI. In the AKI-group, patients were older and showed higher Acute Physiology and Chronic Health Evaluation II (APACHE II) scores at admission, higher rates of sedation, mechanical ventilation, and infection. More drugs in general and specifically more vasoactive drugs were prescribed for AKI group. Patients who developed AKI tended to have extended stays in the ICU and a lower probability of being discharged alive than patients with no AKI development. Model adjustments of logistic regression showed that the number of medications (OR 1.15; 95% CI [1.05–1.27]) was the only factor associated with AKI in this study. This association was independent of drug nephrotoxicity.

**Discussion**. Intensive care is characterized by its complexity that combines unstable patients, severe diseases, high density of medical interventions, and drug use. We found that typical risk factors for AKI showed statistical association on bivariate analysis. The contribution of drug therapy in the occurrence of AKI in medical ICUs reinforces the need for prevention strategies focused on early recognition of renal dysfunction and

interventions in drug therapy. These actions would help improve the quality of patient care and ensure progress towards medication safety.

## INTRODUCTION

Acute kidney injury (AKI) refers to a sudden decline in renal function, often secondary to an injury that leads to structural or functional changes in the kidneys (*Mehta et al., 2007*). It is commonly associated with increased cost and extended hospital stay due to the increased morbidity and mortality associated with it (*Chertow et al., 2005*; *Wang et al., 2012*). AKI was reported to occur in 7–22.7% of hospitalized patients (*Wang et al., 2012*; *Jones & Devonald, 2013*; *Sawhney & Fraser, 2017*) and this range may extend to 19–67% among critically ill patients (*Mataloun et al., 2006*; *Herrera-Gutiérrez et al., 2013*; *Jones & Devonald, 2013*). In intensive care units (ICU), AKI is commonly multifactorial and frequently involves diverse factors such as hypovolemia, sepsis, drugs, hemodynamic disturbances, renal hypoperfusion, intrinsic kidney damage, and post-renal obstruction (*Taber & Mueller, 2006*; *Pannu & Nadim, 2008*; *Dennen, Douglas & Anderson, 2010*).

The conceptual models and definitions for AKI and chronic kidney disease (CKD) have many similarities. Both conditions are characterized primarily by decreased kidney function, which can lead to kidney failure and death, as well as fatal and nonfatal complications in other organ systems. However, the time of onset and duration of functional abnormality is important to distinguish AKI from CKD (*Chawla et al., 2017*). The Kidney Disease: Improve Global Outcomes (KDIGO) has defined AKI as an abrupt decrease in kidney function that occurs over a period of seven days or less, and CKD as abnormalities in kidney structure or function that persist for >90 days (*KDIGO-CKD Group, 2012*). These definitions are useful for routine employment for the diagnosis of these conditions in clinical practice and research (*Chawla et al., 2017*). Patient care should adopt procedures to minimize drug-induced harm related to the use of nephrotoxic drugs, and provide supportive care (*Pozzoli, Simonini & Manunta, 2018*).

The complexity of conditions affecting critically ill patients often requires the use of a large variety of drugs. Many of them, individually or in combination, may induce renal injury, but it is difficult to identify an isolated cause for AKI. The global contribution of drug-induced renal dysfunction in medical ICUs remains unclear (*Perazella, 2012*; *Finlay et al., 2013*). The use of potentially nephrotoxic drugs may compromise patient safety by inducing adverse events involving renal function. In critically ill patients, nephrotoxic drugs were reported to contribute to 14–21% of cases of AKI (*Bernieh et al., 2004*; *Mehta et al., 2004*; *Uchino et al., 2005*; *KDIGO-AKI Group, 2012*). KDIGO guidelines have stated that rates may range from 20–30% (*KDIGO-AKI Group, 2012*). Proposed mechanisms of AKI include direct cell toxicity or a reduction in the renal perfusion, and the severity also depends on patient's clinical conditions (*Mehta et al., 2004*). Regarding all these aspects,

in the recent years significant efforts have been made in the development of new tools, predictive models and biomarkers of AKI (*Pozzoli, Simonini & Manunta, 2018*).

The role of drug therapy in the development of kidney damage is relevant, but often ignored as a preventable cause of AKI and has not been clearly explored in previous studies thus far (*Falconnier et al., 2001*; *Perazella, 2012*; *Finlay et al., 2013*). Temporal and regional assessments of AKI incidence may be influenced by changes in AKI awareness, recognition and clinical practice (*Sawhney & Fraser, 2017*). Risk factors associated with AKI in ICU patients is an important topic that deserves further investigation. Thus, this study was designed to examine drug therapy and other factors associated with the development of AKI in critically ill patients at a tertiary public hospital in Brazil.

## MATERIAL & METHODS

### Setting and design

This is a cross-sectional study involving patients hospitalized in a 35-bed medical ICU of a 312-bed tertiary teaching hospital in Belo Horizonte, Southeast Brazil. This is a referral public hospital focused on clinical and surgical emergency assistance. Standard operating procedures and evidence-based protocols are adopted to provide patient care. Patients admitted to ICU were recruited consecutively from October 1st 2014 to February 27th 2015. The study was approved by the Institutional Review Board on Research Ethics (approval code CAAE 25655014.1.0000.5149) of Universidade Federal de Minas Gerais. We obtained a signed consent from conscious patients or from relatives of those who were sedated and intubated. Patients' relatives who provided informed consent were also their legally authorized representatives. The primary outcome of this study was the incidence of AKI in the ICU patients, and sociodemographics, clinical data and drug therapy information were considered as covariates.

### Study population

The inclusion criteria were: age ≥18 years, no records of kidney disease, more than 24 h since admission to ICU, and total hospitalization time before admission to ICU not higher than seven days. To assess previous kidney disease, we considered records of CKD in medical charts, chronic dialysis or significant increase in serum creatinine ($S_{Cr}$) after hospital admission (≥50%). The reason to include patients with a short hospital stay was to assess patients who had been exposed to less hospital interventions that could affect renal function and serve as nonmeasured confounding factors in the study analyses. For readmissions, only data for the first admission were considered in this study. Patients were excluded if they had records of kidney disease, AKI on admission to ICU, and renal replacement therapy (RRT) prior to hospitalization or ICU admission.

### Data collection

All patients admitted to the ICU during the study period were assessed for potential eligibility on the second day of admission. Data were collected until ICU discharge. Electronic medical charts were reviewed using a pre-tested questionnaire to collect sociodemographic, clinical, and laboratory data. Attending ICU physicians were blinded to
detailed information on the study aims and to the list of patients who had been recruited to participate in this investigation. Patients were characterized by sex, age, race, history of alcohol consumption, and smoking status. Clinical information covered the original hospital department in which the patient was admitted, cause of admission, comorbidities, presence of trauma, sedation, mechanical ventilation, infection and/or sepsis when admitted in the ICU, Acute Physiology and Chronic Health Evaluation II (APACHE II) score and length of stay exclusively at ICU. The need of RRT and evolution to discharge or death was also registered.

Regarding renal function, all data on sequential $S_{Cr}$ level, glomerular filtration rate (GFR), and urine output were collected during ICU stay. The laboratory kit used to measure $S_{Cr}$ was VITROS® (Chemistry Products CREA Slides, Ortho-Clinical Diagnostics, Johnson & Johnson, Raritan, NJ, USA), which is traceable by isotope dilution-liquid chromatography mass spectrometry (ID-LC/MS). The GFR was determined according to the CKD-EPI (Chronic Kidney Disease Epidemiology Collaboration) formula (*Levey et al., 2009*). The Acute Kidney Injury Network (AKIN) criterion (*Mehta et al., 2007*) was used to determine the presence of AKI, using as baseline the first $S_{Cr}$ recorded upon hospital admission. The reason for choosing this first $S_{Cr}$ value during hospitalization was due to the lack of access to patients' medical history prior to hospital admission. Patients were classified to be at the worst stage when they reached their highest recorded $S_{Cr}$ value during their ICU stay. The first, the highest, and the last $S_{Cr}$ measurements at ICU were considered in the final analysis. For patients discharged alive, we described the need for additional treatment due to AKI or the status of recovery (partial or total) from AKI. Status of recovery was based on $S_{Cr}$ levels: partial recovery was the reduction of $S_{Cr}$ levels at discharge, but they were still considered elevated when compared to baseline levels; total recovery consisted on the reduction of $S_{Cr}$ levels to baseline levels detected at hospital admission.

The number of medications prescribed per patient during the ICU stay was extracted from the hospital computerized system. Drugs prescribed in the ICU were classified according to the first level of the Anatomical Therapeutic Chemical (ATC) System (*ATC, 2013*). Vasoactive and potentially nephrotoxic drugs were listed and analyzed as quantitative variables. Vasoactive drugs encompassed dobutamine, dopamine, epinephrine, nitroglycerin, norepinephrine, and sodium nitroprusside. Owing to the heterogeneity of classification of drug-induced nephrotoxicity in information sources (*Bicalho et al., 2015*), a drug was considered nephrotoxic when cited by at least one out of the four selected drug information sources: *Micromedex® (2014)*, *Medscape® (2014)*, *UpToDate® (2014)*, and the National Therapeutic Formulary (NTF) (*National Therapeutic Formulary, 2010*), which is a print publication that is edited by the Brazilian Ministry of Health. The classification of the potential nephrotoxicity of a drug was based on the definition proposed by *Finlay et al. (2013)*. Based on this list, a comprehensive assessment of drug-induced nephrotoxicity and dose adjustment recommendations across drug information sources by drug classes was provided elsewhere (*Bicalho et al., 2015*). Drugs administered by topical and ophthalmic routes, germicides, inhaled drugs, dialysis solutions, and other non-drug products were excluded from this classification.

## Statistical analysis

Baseline patient characteristics were registered by double entry using EpiData software (ver. 3.1; EpiData Assoc, Odense M, Denmark). All data were analyzed with the STATA software (StataCorp. 2013. Stata Statistical Software: Release 13. College Station, TX: StataCorp LP). Descriptive statistical methods were applied using measurements of central tendency and variability, according to the distribution of variables. The presence or absence of AKI was used to classify patients into two groups that were compared using bivariate analysis. The assumption of normal distribution for the continuous variables was assessed by the Shapiro–Wilk test. Chi-square and Mann–Whitney tests were used in the non-adjusted analyses to examine differences between frequencies and medians, respectively. The linear correlation between the number of drugs prescribed at ICU and APACHE II was tested by the Spearman's coefficient with significance level of 5%.

Factors associated with AKI were identified by model adjustments of logistic regression. Independent variables with $p$ values <0.25 in the bivariate analysis were tested in logistic regression models using blocks of variables. The rationale for using $p$-value cut-off point of 0.25 is testing variables at some arbitrary level to help selecting candidates for the multivariate analysis that could not be identified as important variables using more conservative levels, such as 0.05 (*Bursac et al., 2008*). After adjusting the regression model for each block of covariates, the selected variables were considered in the final model of multiple logistic regression (Enter method). The estimated odds ratios (OR) were presented with a 95% confidence interval (CI). All p values were 2-tailed ($\alpha = 0.05$). The variance inflation factor (VIF) was used to identify the presence of multicollinearity (if VIF>10.00). The model was assessed for its goodness-of-fit using the Hosmer–Lemeshow test. The predictive ability of the model was calculated by the estimates for the area under the receiver operating characteristic (ROC) curve. We also performed post-hoc power calculation for sample size.

## RESULTS

During the study, the 296 patients admitted at the ICU were assessed for eligibility and 122 were included for follow-up. The flowchart for patient selection and follow-up until ICU discharge is presented in Fig. 1.

Most patients were men (58.2%) with ages ranging from 18–88 years (median 46.0 years; IQ (interquartile range) = 29.0–69.0). The surgical department (46.7%) and the emergency department (42.6%) handled most of the hospital admissions. The top two causes of hospital admissions were polytrauma (17.2%) and cerebrovascular accident (12.3%). Most patients had at least one comorbidity at admission (58.2%), represented mainly by diabetes (19.7%), cerebrovascular disease (16.4%), and chronic pulmonary disease (15.6%). Twenty six (21.3%) patients had three or more comorbidities (Table 1).

The number of drugs prescribed per patient throughout the ICU stay varied from seven to 51 (median 20.0; IQ = 15.0–27.0). The number of potentially nephrotoxic drugs ranged from two to 24 per patient with a mean of 9.3; SD (standard deviation) = 4.6. A total of 83 (68.0%) patients received at least one vasoactive drug. All drugs in use were

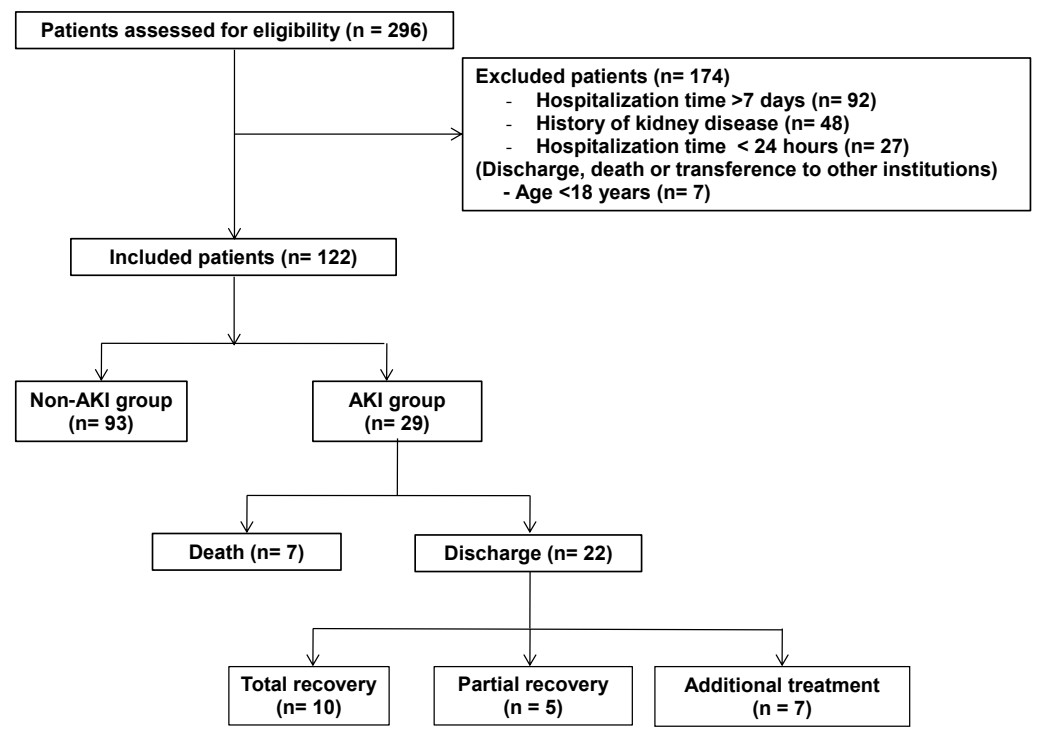

**Figure 1 Flowchart for patient selection and follow-up until ICU discharge.** Abbreviations: AKI, acute kidney injury.

classified according to the first level of ATC showing that the main anatomical groups were "Nervous System" (24.3%), "Alimentary tract and metabolism" (21.0%) and "Blood and blood forming organs" (18.0%) (Table 2). According to the list of potentially nephrotoxic drugs, fentanyl was the most frequently used drug (86; 70.5%). Nephrotoxic antibiotics were represented by vancomycin (37; 30.3%), ceftriaxone (31; 25.4%), polymyxin (31; 25.4%), cefazolin (15; 12.0%), amikacin (13; 10.7%) and gentamicin (9; 7.4%) (Table 3).

Figure 2 depicts the three chosen values of $S_{Cr}$ (value at admission, the highest value during the ICU stay and the last $S_{Cr}$) and the estimated GFR for the groups with or without AKI. In the AKI group, $S_{Cr}$ was higher and estimated GFR was lower than the values presented by the non-AKI group.

According to AKIN criterion, 29 (23.8%) patients developed AKI. From these, 10 (34.5%) required hemodialysis at some point of their ICU stay and seven (24.1%) reached the AKIN 3 stage. Seven out of 11 deaths corresponded to the AKI group. After discharge, seven patients required interventions due to AKI developed during ICU stay (clinical follow-up or RRT), five had partial recovery, and 10 had total recovery from AKI (Fig. 1).

Sociodemographic and clinical data were compared between both groups by bivariate analysis, which is also shown in Table 1. In the AKI-group, patients were older and showed higher APACHE II scores at admission, higher rates of sedation, mechanical ventilation, and infection. More drugs in general and specifically more vasoactive drugs were prescribed for AKI group. Additionally, the group of patients who developed AKI tended to have extended

**Table 1  Bivariate analysis of sociodemographic, clinical, and medication data in AKI and non-AKI-groups of patients admitted at a teaching hospital ICU.**

| Variables | Total (*n* = 122) | Non-AKI group (*n* = 93) | AKI group (*n* = 29) | *p*-value[a, b] |
|---|---|---|---|---|
| *Sociodemographic data* | | | | |
| Female sex, *n* (%) | 51 (41.8) | 39 (41.9) | 12 (41.4) | 0.96[a] |
| Age, median (IQ) | 46.0 (29.0–69.0) | 41 (28.0–63.0) | 62.0 (46.0–76.0) | 0.01[b] |
| Race, *n* (%) | | | | |
| White | 95 (77.9) | 76 (81.7) | 19 (65.5) | 0.07[a] |
| Non-white | 27 (22.1) | 17 (18.3) | 10 (34.5) | |
| History of ethanol use, *n* (%) | 45 (37.8) | 34 (37.8) | 11 (37.9) | 0.99[a] |
| Smoking status, *n* (%) | 35 (29.4) | 26 (28.9) | 9 (31.0) | 0.83[a] |
| *Clinical data* | | | | |
| APACHE II Score at admission ICU, (IQ) | 21.4 (15.0–27.0) | 20.1 (14.0–25.0) | 25.6 (20.0–31.0) | <0.01[b] |
| Comorbidities, *n* (%) | | | | |
| 0 | 51 (41.8) | 44 (47.3) | 7 (24.1) | 0.05[a] |
| 1–2 | 45 (36.9) | 33 (35.5) | 12 (41.4) | |
| >2 | 26 (21.3) | 16 (17.2) | 10 (34.5) | |
| Sedation, *n* (%) | 67 (54.9) | 45 (48.4) | 22 (75.9) | 0.01[a] |
| Trauma, *n* (%) | 47 (38.5) | 40 (43.0) | 7 (24.1) | 0.07[a] |
| Mechanical ventilation, *n* (%) | 82 (67.2) | 58 (62.4) | 24 (82.8) | 0.04[a] |
| Infection, *n* (%) | 49 (40.2) | 29 (31.2) | 20 (69.0) | <0.001[a] |
| Sepsis, *n* (%) | 20 (16.4) | 12 (12.9) | 8 (27.6) | 0.06[a] |
| Length of ICU stay[c] (days), *n* (%) | | | | |
| Up to 5 | 56 (45.9) | 50 (53.8) | 6 (20.7) | 0.01[a] |
| From 6 to 15 | 36 (29.5) | 23 (24.7) | 13 (44.8) | |
| More than 15 | 30 (24.6) | 20 (21.5) | 10 (34.5) | |
| ICU Discharge, *n* (%) | | | | |
| Discharge | 111 (91.0) | 89 (95.7) | 22 (75.9) | <0.01[a] |
| Death | 11 (9.0) | 4 (4.3) | 7 (24.1) | |
| *Medications* | | | | |
| Number of drugs in general, median (IQ) | 20.0 (15.0–27.0) | 19.0 (14.0–26.0) | 25.0 (20.0–35.0) | <0.001[b] |
| Number of potentially nephrotoxic drugs, *n* (%) | | | | |
| 0–6 | 34 (27.9) | 30 (32.3) | 4 (13.8) | 0.06[a] |
| 7–11 | 55 (45.1) | 42 (45.2) | 13 (44.8) | |
| >11 | 33 (27.1) | 21 (22.6) | 12 (41.4) | |
| Number of vasoactive drugs, *n* (%) | | | | |
| 0 | 39 (32.0) | 35 (37.6) | 4 (13.8) | <0.01[a] |
| 1–2 | 75 (61.5) | 55 (59.1) | 20 (69.0) | |
| >2 | 8 (6.5) | 3 (3.2) | 5 (17.2) | |

**Notes.**

Abbreviations: AKI, acute kidney injury; ICU, intensive care unit; IQ, interquartile range; APACHE II, Acute Physiology and Chronic Health Evaluation II; ICU, intensive care unit.

[a] $\chi^2$ de Pearson.

[b] Mann-Whitney.

[c] Length of hospital stay prior to ICU admission was not included in this analysis.

**Table 2** Drug classification according to the first level of ATC.

| Anatomical groups | Frequency, *n* (%) |
|---|---|
| N: Nervous system | 658 (24.3) |
| A: Alimentary tract and metabolism | 569 (21.0) |
| B: Blood and blood forming organs | 487 (18.0) |
| C: Cardiovascular system | 396 (14.6) |
| J: Antiinfectives for systemic use | 313 (11.6) |
| m: Musculo-skeletal system | 84 (3.1) |
| R: Respiratory system | 80 (3.0) |
| H: Systemic hormonal preparations, excl. sex hormones and insulins | 56 (2.1) |
| D: Dermatologicals | 26 (1.0) |
| V: Various | 23 (0.9) |
| S: Sensory organs | 10 (0.4) |
| P: Antiparasitic products, insecticides and repellents | 2 (0.1) |

**Notes.**
Abbreviations: ATC, Anatomical Therapeutic Chemical Classification.

stays in the ICU and a lower probability of being discharged alive than the non-AKI group. Our analysis showed a significant correlation between the number of drugs prescribed at ICU and APACHE II (Spearman's coefficient 0.56; $p < 0.001$).

After adjusting the logistic regression model, the variables maintained in the final model were age, race, number of comorbidities, length of ICU stay and number of prescribed drugs. Only the total number of drugs was significantly associated with AKI (OR 1.15; 95% CI [1.05–1.26]) (Table 4). We did not identify multicollinearity for these variables indicated by VIF values < 3.00. The final model fitted the data well (Hosmer–Lemeshow test = 0.9397) and presented a good predictive ability (area under the ROC curve = 0.8102) (Fig. 3). Figure 4 shows that increase in the number of drugs prescribed increased the probability of the occurrence of AKI, independent of the potential of those drugs to induce nephrotoxicity. Based on literature data (*Santos & Monteiro, 2015*), we determined that our statistical power was equal to 71%, with 95% confidence, to identify the difference in the probability of development of renal damage in 39 non-exposed and 83 exposed to vasoactive drugs.

## DISCUSSION

Our results showed that 23.8% of studied patients had AKI during their ICU stay at a Brazilian teaching hospital. Previous studies have reported wide variations in the incidence of AKI in ICUs by using different diagnostic criteria. Two studies found incidences of 25.5% (*Ponce et al., 2011*) and 39.3% (*Nisula et al., 2013*) using AKIN classification. Other authors reported an incidence range of 35.7–47.0% by using the Risk/Injury/Failure/Loss/End-stage (RIFLE) system (*Bagshaw et al., 2008*; *Wahrhaftig, Correia & De Souza, 2012*; *Mohamed et al., 2013*). In a prospective study including 29,269 patients hospitalized at 54 ICUs from 23 countries, *Uchino et al. (2005)* found 1738 patients (5.7%; 95% CI [5.5–6.0]%) who

**Table 3 Frequency of use of potentially nephrotoxic drugs in the ICU patients studied at a teaching hospital (*n* = 122).**

| Potentially nephrotoxic drugs | Frequency[a], *n* (%) | Potentially nephrotoxic drugs | Frequency[a], *n* (%) |
|---|---|---|---|
| Fentanyl | 86 (70.5) | Propofol | 15 (12.3) |
| Morphine | 79 (64.8) | Dobutamine | 14 (11,5) |
| Omeprazole | 78 (63.9) | Amikacin | 13 (10.7) |
| Norepinephrine | 73 (59.8) | Amiodarone | 12 (9.8) |
| Magnesium sulfate | 54 (44.3) | Carvedilol | 12 (9.8) |
| Furosemide | 46 (37.7) | Spironolactone | 12 (9.8) |
| Vancomycin | 37 (30.3) | Amoxicillin + clavulanic acid | 11 (9.0) |
| Diazepam | 35 (28.7) | Ciprofloxacin | 11 (9.0) |
| Simvastatin | 33 (27.1) | Valproic acid | 10 (8.2) |
| Metronidazole | 32 (26.2) | Clopidogrel | 10 (8.2) |
| Ceftriaxone | 31 (25.4) | Non-ionic contrast media | 10 (8.2) |
| Meropenem | 31 (25.4) | Hydralazine | 10 (8.2) |
| Polymyxin | 31 (25.4) | Clarithromycin | 9 (7.4) |
| Clonazepam | 27 (22.1) | Clonidine | 9 (7.4) |
| Phenytoin | 27 (22,1) | Gentamicin | 9 (7.4) |
| Captopril | 25 (20.5) | Calcium gluconate | 9 (7.4) |
| Amlodipine | 24 (19.7) | Phenobarbital | 8 (6.6) |
| Risperidone | 19 (15.6) | Clindamycin | 7 (5.8) |
| Losartan | 17 (13.9) | Epinephrine | 7 (5.8) |
| Sodium nitroprusside | 17 (13.9) | Hydrochlorotiazide | 7 (5.7) |
| Acetaminophen | 17 (13.9) | Oxacillin | 7 (5.7) |
| Piperacillin + tazobactam | 17 (13.9) | Atropine | 6 (4.9) |
| Tramadol | 17 (13.9) | Cefepime | 6 (4.9) |
| Cefazolin | 15 (12.3) | | |

**Notes.**
[a] Potentially nephrotoxic drugs with frequency of use <5 patients were not shown in this table.
[b] Iohexol or iopamidol or ioversol or iobitriol.

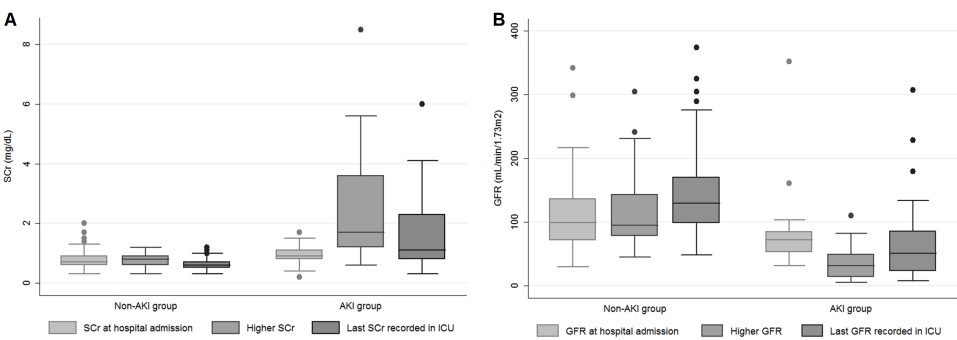

**Figure 2** $S_{Cr}$ **(A) and GFR (B) at three different moments of hospitalization according to AKI and non-AKI groups.** Abbreviations: $S_{Cr}$, serum creatinine; AKI, acute kidney injury; GFR, glomerular filtration rate.

**Table 4** **Final model of logistic regression for the factors associated with acute kidney injury in intensive care unit patients of a teaching hospital.**

| Variables | Odds Ratio | 95% CI |
|---|---|---|
| Age | 1.02 | 0.99–1.05 |
| Race | | |
|     Non-white | 1.00 | – |
|     White | 1.28 | 0.40–4.10 |
| Comorbidities | | |
|     0 | 1.00 | – |
|     1 –2 | 1.29 | 0.34–4.93 |
|     >2 | 0.95 | 0.20–4.45 |
| Length of ICU stay[a] (days) | | |
|     Up to 5 days | 1.00 | – |
|     From 6 to 15 | 1.64 | 0.42–6.39 |
|     More than 15 | 0.24 | 0.28–2.02 |
| Number of drugs in general | 1.15 | 1.05–1.27 |
| $S_{Cr}$ at hospital admission | 2.83 | 0.25–31.60 |
| GFR at hospital admission | 1.00 | 0.98–1.01 |
| APACHE II at admission ICU | 0.99 | 0.93–1.06 |

**Notes.**

Abbreviations: AKI, acute kidney injury; ICU, intensive care unit; CI, confidence interval; $S_{Cr}$, serum creatinine; GFR, glomerular filtration rate; APACHE II, Acute Physiology and Chronic Health Evaluation II (APACHE II).

[a]Length of hospital stay prior to ICU admission was not included in this analysis.

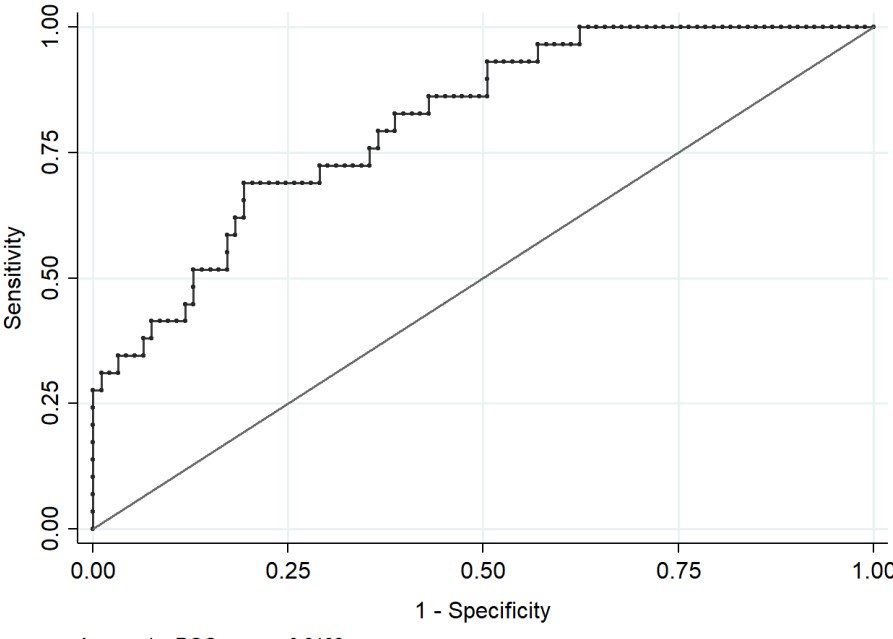

Area under ROC curve = 0.8102

**Figure 3** **ROC curve for the final model.**

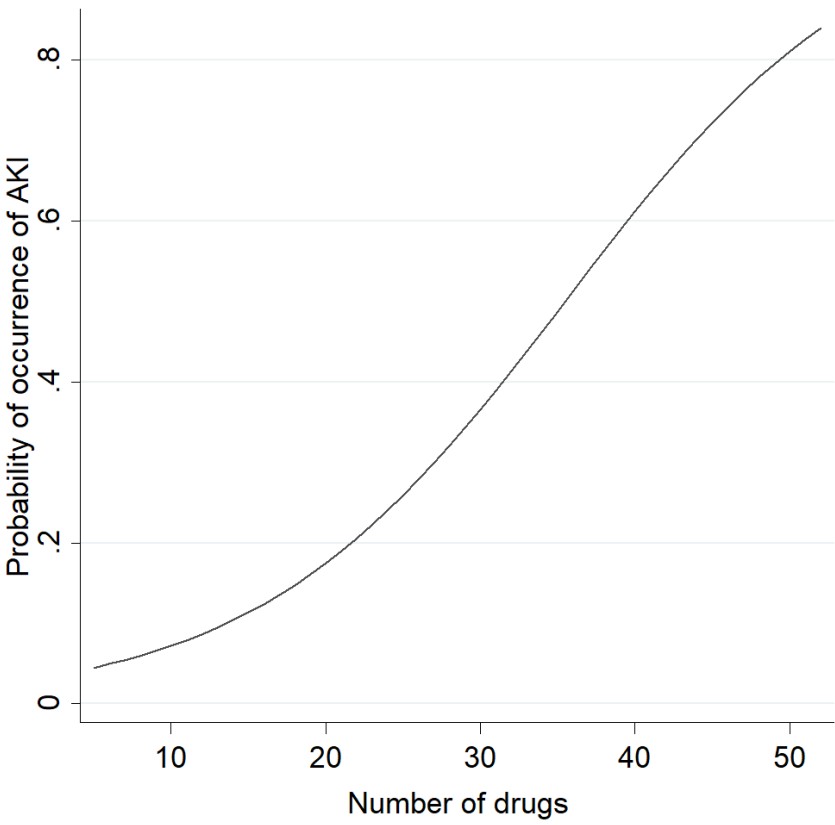

**Figure 4** **Probability of occurrence of AKI according to the number of drugs used by the studied ICU patients.** Abbreviations: AKI, acute kidney injury; ICU, intensive care unit.

had AKI some time during their ICU stay, with a frequency ranging from 1.4–25.9% of all patients.

Bivariate analysis highlighted the relevant risk factors for AKI that have been investigated individually or in combination by other observational studies (*Mataloun et al., 2006*; *Prakash et al., 2006*; *Taber & Mueller, 2006*; *Bagshaw et al., 2008*; *Perazella, 2012*; *Wahrhaftig, Correia & De Souza, 2012*; *Wang et al., 2012*; *Herrera-Gutiérrez et al., 2013*; *Mohamed et al., 2013*). We found that typical risk factors were associated with AKI in the studied patients, including advanced age, higher APACHE II score at admission to ICU, sedation, mechanical ventilation, and infection. Clinically, patients presenting these characteristics show a high degree of frailty to develop AKI.

Previous studies have discussed the contribution of drugs to the occurrence of AKI in critically ill patients. In the prospective study mentioned earlier, drug-induced nephrotoxicity was a contributing factor identified in 19.0% of AKI patients (*Uchino et al., 2005*). *Bernieh et al. (2004)* retrospectively evaluated 81 patients in a tertiary hospital reporting that in 11 patients (14%), the exposure to nephrotoxic drugs contributed to the occurrence of AKI. In the present study, age, race, number of comorbidities, length of ICU stay, number of drugs, $S_{Cr}$ level, GFR, and APACHE II score at hospital admission were

the variables included in the final multiple logistic regression model, but only the number of drugs showed significant statistical association with AKI (OR 1.15; 95% CI [1.05–1.27]). Interestingly, nephrotoxicity of the drugs did not appear to be a factor associated with AKI. Certain inferences regarding drug-induced AKI in medical ICUs require further clarification in future studies that should provide clearer definitions for the assessed nephrotoxic drugs and appropriate adjustments for residual confounding variables. We acknowledge that a global understanding of the development of AKI should take into account the influence of individual and concomitant nephrotoxins, their proposed mechanism of nephrotoxicity, dose, drug potency for toxicity and the duration of drug therapy.

Regarding studies developed in Brazil, *Reis & Cassiani (2011)* conducted a cross-sectional study with 299 medical records of patients admitted for ≥5 days in an ICU. Drug-induced AKI was the most frequent (22; 16.3%) adverse drug event in the studied hospital. Other study assessed 524 ICU patients to investigate clinical aspects related to acute tubular necrosis classified as ischemic, mixed and nephrotoxic. There was a low incidence (58/524) of isolated nephrotoxic acute tubular necrosis, defined as exposure to nephrotoxins 72 h preceding the increase of $S_{Cr}$ (*Santos et al., 2006*). In our study, the most frequently used nephrotoxic antibiotics were vancomycin (37; 30.3%), polymyxin (31; 25.4%) and ceftriaxone (31; 25.4%). *Mostardeiro et al. (2013)* analyzed nephrotoxicity rates in 92 solid organ transplant patients who had taken polymyxin, reporting rates of 25%, 30%, and 51% on days 9, 16, and 29, respectively, after starting polymyxin. An independent association between duration of polymyxin use and AKI has been reported ($p = 0.037$; OR 1.06; 95% CI [1.00–1.12]). With respect to the combination of nephrotoxins, *Soares et al. (2017)* investigated the incidence of AKI in 115 ICU patients who used polymyxin B plus vancomycin (Group I) or polymyxin B alone (Group II), showing higher incidence of AKI in Group I than in Group II (62.7% vs. 28.5%, $p = 0.005$). Due to the heterogeneity of studies, there are some limitations for comparison among studies. However, we could state that strategies focused on close monitoring of patients taking nephrotoxic drugs alone or in combination would be relevant to minimize the risks for AKI development. In our study, other important nephrotoxins had a low frequency of use, such as aminoglycosides and contrast media.

Individual drugs may have significant pharmacokinetic alterations in critically ill patients, especially for drugs with a small volume of distribution or a high percentage of binding to plasma proteins. Enteral absorption can be reduced owing to the increase in gastric pH, low intestinal motility, drug-food interactions, and intestinal edema. Depending on drug properties, drug distribution may be altered as a result of changes in total body water, blood supply, concentration of plasma proteins and binding affinity, tissue permeability, and pH of biological fluids (*Schetz et al., 2005*; *Perazella, 2012*). Drug-induced nephrotoxicity may cause additional damage related to hemodynamic alterations and direct injury to tubular cells (*Perazella & Setaro, 2003*). Vasoactive drugs were used by 68.0% of studied patients. Higher doses of these agents are commonly indicated in critically ill patients to help in adjusting blood pressure by vasoconstriction. However, they may also

reduce renal blood flow and cause renal hypoxia and acute tubular necrosis with prolonged use (*Taber & Mueller, 2006*). Vasoactive drugs did not appear as an independent factor for AKI in our study.

Intensive care is characterized by a complex environment that combines unstable patients, severe diseases, high density of medical interventions, and drug use (*Moyen, Camiré & Stelfox, 2008*). Critically ill patients are at high risk for adverse drug events because of the severity of disease, organ dysfunction, and the number, complexity, and duration of medications administered (*Manenti et al., 1998*; *Padilha et al., 2002*; *Perazella, 2012*; *Preslaski et al., 2013*). The reason for ICU admission might also contribute to clinical complications, such as AKI. In a prospective intervention study including 138 patients in an ICU in Germany, 68 (49%) patients had renal dysfunction, and 14% (110/805) of prescribed drugs required consideration of renal function. A potential overdose was found in 48% (53/110) of drugs and this percentage was reduced to 24% ($p < 0.001$) after the intervention (*Bertsche et al., 2009*). Inappropriate doses in patients with renal dysfunction may lead to clinical complications, extended hospital stay and avoidable excess costs. Individualized adjustments of drug doses are essential to maximize therapeutic effectiveness and to ensure patient safety (*Falconnier et al., 2001*). In this context, the expansion of critical care pharmacy services was stated to be relevant to improve patient outcomes and the quality of care. The inclusion of a clinical pharmacist as a member of the multidisciplinary ICU team was reported to protect patients from drug-induced renal impairment (*Preslaski et al., 2013*).

The literature displays a paucity of high quality evidence from prospective studies investigating the impact of the use of nephrotoxic drugs and the occurrence of AKI in critical care. Investigations on correct dosing of nephrotoxic or other drugs in patients with AKI would be of clinical relevance. Future studies on this topic should also address the performance of systems for the early detection of drug-induced AKI and the use of more accurate protocols to improve the management of drug regimens and consequently prevent the expansion of renal injury in patients. Serum and urinary biomarkers in monitoring are promising and exhibit prominent advantages, although their use in clinical practice is still limited (*Ferguson, Vaidya & Bonventre, 2008*; *Peres et al., 2013*). The present study provided a better understanding of the occurrence of AKI in a Brazilian healthcare scenario and highlighted the number of prescribed medications throughout ICU stay as an important risk factor for this complication. This hypothesis needs further investigation in large multicenter studies designed to address drug-induced AKI.

Limitations of this study deserve to be mentioned. First, the use of $S_{Cr}$ level as a biomarker for estimating renal function and the eventual inconsistencies in diuresis measurements found on hospital records may introduce some imprecision to data collection and to the employment of AKIN criterion. Although $S_{Cr}$ levels have been used as the mainstay for the diagnosis of AKI, they do not accurately reflect GFR. Sensitivity and specificity of $S_{Cr}$ measurement are questionable due to the lack of a temporal relationship in providing alterations immediately after the establishment of renal injury, a fact that probably delays AKI diagnosis. Thus, the incidence of AKI reported herein may be underestimated. Second, the ideal baseline $S_{Cr}$ to assess renal function would be a recent value provided

by a family doctor. Taking the first $S_{Cr}$ on admission - and not a true baseline $S_{Cr}$ - could also underestimate the incidence of AKI since many patients may have had an elevation of $S_{Cr}$ upon admission and less variation along the hospitalization precluding the identification of AKI under the AKIN classification. Third, the exclusion of patients with CKD could have interfered in the rates of AKI in this study. Fourth, we do not exclude the influence of unmeasured clinical conditions as contributing factors to AKI, such as delirium, hypotension prior to ICU admission and dosing regimen of prior nephrotoxin use (e.g., long-term use of non-steroidal drugs). Fourth, as the highest AKI stage and the number of drugs were measured throughout the entire ICU stay, it was not possible to draw conclusions on the direction of associations which is a limitation inherent to cross-sectional studies. Finally, the constant changes in ICU prescriptions may have caused an overestimation of the number of medications, since prescribed drugs could have been suspended prior to administration.

## CONCLUSIONS

We found that typical risk factors for AKI showed statistical association on bivariate analysis, including advanced age, higher APACHE II score at admission, sedation, mechanical ventilation, and infection. The number of drugs itself was the only factor independently associated with AKI in the critically ill patients under study, independent of drug nephrotoxicity. The contribution of drug therapy in the occurrence of AKI in medical ICUs reinforces the need for prevention strategies focused on early recognition of renal dysfunction and interventions in drug therapy. These actions would help improve the quality of patient care and ensure progress towards medication safety.

### Funding

The authors received funding from Pró-Reitoria de Pesquisa da Universidade Federal de Minas Gerais for the professional English editing of the manuscript. The funders had no role in study design, data collection and analysis, decision to publish, or preparation of the manuscript.

### Grant Disclosures

The following grant information was disclosed by the authors:
Pró-Reitoria de Pesquisa da Universidade Federal de Minas Gerais.

### Competing Interests

The authors declare there are no competing interests.

### Author Contributions

- Danielly Botelho Soares performed the experiments, prepared figures and/or tables, authored or reviewed drafts of the paper, approved the final draft.

- Juliana Vaz de Melo Mambrini analyzed the data, prepared figures and/or tables, approved the final draft.
- Gabriela Rebouças Botelho and Flávia Fialho Girundi performed the experiments, approved the final draft.
- Fernando Antonio Botoni conceived and designed the experiments, contributed reagents/materials/analysis tools, approved the final draft.
- Maria Auxiliadora Parreiras Martins conceived and designed the experiments, prepared figures and/or tables, authored or reviewed drafts of the paper, approved the final draft, critically reviewed and assisted with the editing of the manuscript.

## Human Ethics

The following information was supplied relating to ethical approvals (i.e., approving body and any reference numbers):

The study was approved by the Review Board on Research Ethics of the Universidade Federal de Minas Gerais (approval code CAAE 25655014.1.0000.5149).

## Data Availability

The raw data are provided in a Supplemental File.

## Supplemental Information

Supplemental information for this article can be found online at http://dx.doi.org/10.7717/peerj.5405#supplemental-information.

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
