# Peer review of "Drug therapy and other factors associated with the development of acute kidney injury in critically ill patients: a cross-sectional study"

_PeerJ, doi:10.7717/peerj.5405_

## Round 0.1 · original submission · Minor Revisions

All issues proposed by two reviewers should be resolved before further consideration.

·

Basic reporting

This is an interesting article about acute kidney injury (AKI) in an intensive care unit of a tertiary hospital in a large city in southeast region of Brazil. The main finding was that multi-drug therapy was the main risk factor for AKI development, which often includes nephrotoxic drugs, and physicians must be aware of this risk. The text is well written and the data presented is relevant. I suggest some minor corrections before publication.

Experimental design

The study design requires to be better detailed because it is a little confusing, for example, it was not clear how the authors have distinguished a patient as having AKI or chronic kidney disease, and which criteria was used for considering complete or partial renal fucntion recovery at the time of ICU or hospital discharge. I also wonder if the study can be considered as cross-sectional or longitudinal (if patients were followed until the time of hospital discharge or only at the time of ICU admission or until the time of AKI development). A flowchart describing all the methodological process would clarify this questions, showing all the moments of the study.

Validity of the findings

In the results it would be interesting to provide more details about the drugs that were used in these group of patients (not only the number of drugs). The reader will certainly have the curiosity to know which drugs were most often used, and which were the potential nephrotoxic drugs. Maybe a table showing all nephrotoxic drugs used, with its frequency would add important information in the results section.
In the discussion section I would suggest to include more studies from Brazil because thera are some important studies of AKI in ICU in that country that should be cited and compared with the results of this paper. I also suggest to discuss a little bit more in detail about some specific important nephrotoxic drugs, such as amphotericin B, vancomycin, polymyxin, once they are frequently used in clinical practice and have important impact on AKI development.

Additional comments

The paper is, in general, well written, the findings are interesting and deserves publication. I suggest to give more details about some specific nephrotoxic drugs and try to compare with other studies in Brazil, once there are some studies in this area in this country that should be cited.

·

Basic reporting

no comment

Experimental design

no comment

Validity of the findings

no comment

Additional comments

Overall, I found the manuscript to be well-written and the study design and methods were clearly defined. Some specific comments for consideration are included below:
1. Line 66, the sentence beginning “However, the period of functional…” needs to be expounded upon. What is the cutoff period, etc.?
2. Line 72, not sure that treatments is the correct word to be used.
3. The final sentence of paragraph 2 of the introduction starting on line 73; the previous statements do not justify this conclusion. It would probably be better placed after the next paragraph.
4. Line 82, nephrotoxic or “potentially” nephrotoxic drugs. Clarify since it would be unwise to give critically ill patients drugs known to cause nephrotoxicity.
5. Line 121, should be “for” instead of “of”.
6. Laboratorial is used twice, line 129 and line 141. This is not the correct usage of this word. Consider “clinical chemistry” or something along that line instead.
7. Line 186, how was p<0.25 chosen for testing in logistic regression models since this is not considered a significant value.
8. Lines 238-239, it would be good to show the ROC curve for the model when mentioned.
9. Line 304, should be “for” instead of “of”.
10. For the dataset, the headings for the columns should be defined at the end of the dataset or beginning.
11. For the dataset, it appears that there are 125 patients but only 122 were used for analysis. Why is there a discrepancy here?

---

## Round 0.2 · accepted · Accept

We think the concerns the reviewers proposed are solved and the manuscript could be accepted for publication. Thank you for the submission.

·

Basic reporting

The authors report interesting findings about acute kidney injury in an intensive care unit in Brazil. The corrections were made according to reviewers suggestions, and there is no any other correction required.

Experimental design

The research question is well defined and relevant and methods are well written and correctly described.

Validity of the findings

The finding are very interesting and conclusiong are valid.

Additional comments

The manuscript can be published.